# Genomic Insights into Cyanide Biodegradation in the *Pseudomonas* Genus

**DOI:** 10.3390/ijms25084456

**Published:** 2024-04-18

**Authors:** Lara P. Sáez, Gema Rodríguez-Caballero, Alfonso Olaya-Abril, Purificación Cabello, Conrado Moreno-Vivián, María Dolores Roldán, Víctor M. Luque-Almagro

**Affiliations:** 1Departamento de Bioquímica y Biología Molecular, Edificio Severo Ochoa, Campus de Rabanales, Universidad de Córdoba, 14071 Córdoba, Spain; bb2samel@uco.es (L.P.S.); a62rocag@uco.es (G.R.-C.); b22olaba@uco.es (A.O.-A.); bb1movic@uco.es (C.M.-V.); bb2rorum@uco.es (M.D.R.); 2Departamento de Botánica, Ecología y Fisiología Vegetal, Edificio Celestino Mutis, Campus de Rabanales, Universidad de Córdoba, 14071 Córdoba, Spain; bv1cahap@uco.es

**Keywords:** comparative genomic, cyanide, nitrilase, *Pseudomonas*, pan-genome

## Abstract

Molecular studies about cyanide biodegradation have been mainly focused on the hydrolytic pathways catalyzed by the cyanide dihydratase CynD or the nitrilase NitC. In some *Pseudomonas* strains, the assimilation of cyanide has been linked to NitC, such as the cyanotrophic model strain *Pseudomonas pseudoalcaligenes* CECT 5344, which has been recently reclassified as *Pseudomonas oleovorans* CECT 5344. In this work, a phylogenomic approach established a more precise taxonomic position of the strain CECT 5344 within the species *P. oleovorans*. Furthermore, a pan-genomic analysis of *P. oleovorans* and other species with cyanotrophic strains, such as *P. fluorescens* and *P. monteilii*, allowed for the comparison and identification of the *cioAB* and *mqoAB* genes involved in cyanide resistance, and the *nitC* and *cynS* genes required for the assimilation of cyanide or cyanate, respectively. While cyanide resistance genes presented a high frequency among the analyzed genomes, genes responsible for cyanide or cyanate assimilation were identified in a considerably lower proportion. According to the results obtained in this work, an in silico approach based on a comparative genomic approach can be considered as an agile strategy for the bioprospection of putative cyanotrophic bacteria and for the identification of new genes putatively involved in cyanide biodegradation.

## 1. Introduction

In recent decades, the Earth has been subjected to such chemical inputs that it could exceed its natural assimilation capacity [1]. In addition to the anthropogenic dispersion of geogenic chemicals as a consequence of mining, farming, and other human activities, synthetic compounds are constantly produced, combined, and released in vast amounts [2]. Thus, pollutants such as plastics are reaching extremely high levels on the planet [3]. These and other persistent organic contaminants have gained a special importance, but traditional inorganic pollutants like heavy metals, nitrate, or cyanide still generate serious environmental problems worldwide [4]. Bioremediation is an ecofriendly technology based on the capacity of bacteria and other organisms to degrade toxic compounds as a result of the great versatility of their metabolic pathways [5]. Bioprospection for novel bacteria with a diverse degradative potential [6] and the designing of genetically engineered bacteria with improved biodegradative capabilities [7] are strategies necessary for the development of new bioremediation approaches. Currently, the availability of huge amounts of bacterial genomes is also allowing us to mine for novel catabolic enzymes and to predict the degradative capacity of bacteria [8,9]. For this purpose, knowledge of the genes required for a specific phenotype and comparative genomic analyses are required. Beyond the environmental application, genomics is also an emerging tool to establish evolutionary relationships among organisms (phylogenomics) in a more precise way than the traditional techniques based on the 16S rRNA or multilocus sequence analyses using a few housekeeping genes [10,11].

Cyanide and cyano-derivatives are a group of inorganic or organic compounds that contain the cyano group (C≡N). Inorganic cyanides include free cyanide (CN¯ and HCN), metal–cyanide complexes (MeCN), and the cyano-derivatives cyanate (OCN¯) and thiocyanate (SCN¯). Organic cyanide forms, also known as nitriles, contain the cyano group bound to an organic skeleton (RCN). Cyanide is naturally produced by different cyanogenic organisms, but the anthropogenic sources are the main input of cyanide into the environment. The extensive use of cyanide in different industrial processes and activities has turned this compound into an essential element for the current welfare of society [12], simultaneously generating a large amount of toxic cyanide-containing wastes that pose a serious threat to the environment and human health.

Cyanide toxicity is determined by the extremely high affinity that this compound displays for metals, making metalloenzymes their main targets. One of these enzymes is the cytochrome *c* oxidase, which is essential for energy production through oxidative phosphorylation in most living organisms. Therefore, some bacteria have developed a resistance to cyanide through the induction of a cyanide-insensitive terminal oxidase acting in cellular respiration [13]. The prokaryotic cyanide-insensitive oxidase (Cio) is a cytochrome *bd*-type quinol oxidase that has been described in different bacteria like *Campylobacter jejuni* and *Gluconobacter oxydans*, as well as in the cyanide-producing bacterium *Pseudomonas aeruginosa* PA01 and the cyanotroph *Pseudomonas pseudoalcaligenes* CECT 5344 [13,14]. Cyanotrophy is the ability of some bacteria to assimilate cyanide as a nitrogen source. Although different cyanide assimilation pathways have been described up to now, including oxidative, hydrolytic, reductive, and substitution/transfer reactions [15,16,17], all of them converge on the formation of ammonia as a final product, which is further incorporated to organic nitrogen by the glutamine synthetase/glutamate synthase (GS/GOGAT) cycle. Cyanide dioxygenase catalyzes the oxidative conversion of cyanide to ammonia and carbon dioxide as final products. The hydrolytic pathways for cyanide assimilation produce formate and ammonia, either directly by the bacterial cyanidase CynD (cyanide dihydratase) or via formamide by the fungal enzyme cyanide hydratase. Nitriles are also assimilated by the hydrolytic enzymes nitrilases and nitrile hydratases, which convert nitriles to ammonia and the corresponding acid or amide, respectively. Nitrogenase has been also described to use cyanide as a substrate, catalyzing the reductive production of ammonia and methane. Finally, the substitution/transfer reactions are catalyzed by the cyanoalanine synthase or the thiosulfate:cyanide sulfurtransferase, which allows cyanide assimilation through the intermediates 3-cyanoalanine or thiosulfate, respectively.

Some *Pseudomonas* strains are able to assimilate cyanide. In *P. pseudoalcaligenes* CECT 5344 and *P. fluorescens* NCIMB 11764, the cyanide assimilation pathway includes the non-enzymatic formation of a 2-hydroxinitrile, which is further hydrolyzed by the nitrilase NitC to produce ammonia and a carboxylic acid [18,19]. The *nitC* gene coding for this nitrilase is located in the *nit1C* gene cluster that also includes genes coding for a member of the *S*-adenosylmethionine superfamily (NitD), an *N*-acetyltransferase enzyme (NitE), a 5-aminoimidizole ribonucleotide synthase (NitF), an NADH-dependent oxidoreductase (NitH), and two proteins of unknown function (NitB and NitG). Upstream, and in the opposite orientation to the genes *nitBCDEFGH*, there is a gene coding for a putative transcriptional regulator (NitA), which belongs to the Fis-family σ^54^-dependent regulators. The *nit1C* gene cluster was initially described as a highly conserved cluster across distant bacterial taxa like *Verrucomicrobium*, *Klebsiella*, *Photorhabdus*, *Burkholderia*, *Rubrivivax*, and *Synechococcus*, among others. Despite the absence of experimental evidence about a putative function of this cluster, a functional connection between genes composing the *nit1C* gene cluster was proposed [20]. In addition to the cyanotrophic strains of the *Pseudomonas* genus in which the cyanide-assimilating capacity has been experimentally linked to the *nit1C* gene cluster, the assimilation of cyanide in *Pseudomonas monteilii* BCN3 was also attributed to the presence of this gene cluster in its genome. Similarly, in a metagenomic analysis performed in Ace Lake (Antarctica), the high coverage of the *nit1C* cluster in the oxic zone was proposed to be associated to the presence of cyanide generated by cyanobacteria, allowing the phylotype of *Candidatus* Regnicoccus frigidus sp. Nov. more abundant in this stratum to assimilate cyanide [21].

In *P. pseudoalcaligenes* CECT 5344, the cyanide assimilation pathway and the cyanide-resistant respiration are processes closely linked through a malate:quinone oxidoreductase (Mqo). Under cyanotrophic conditions, Mqo overproduces oxaloacetate, which chemically reacts with cyanide to produce a cyanohydrin intermediate. In addition, electrons generated during the oxidation of malate are redirected to the cyanide-insensitive oxidase CioAB [14]. In the strain CECT 5344, the *cioAB* genes coding for the alternative oxidase are located in a large cluster that contains 14 genes (BN5_1899-BN5_1912), most of them coding for enzymes putatively involved in the metabolism of several amino acids, like phosphoserine aminotransferase SerC, histidinol-phosphate aminotransferase HisC, acetylornithine aminotransferase ArgD, 4-hydroxy-tetrahydrodipicolinate synthase DapA, methylenetetrahydrofolate reductase MetF, and cysteine synthase CysM. The *cio* cluster also contains genes that code for the sulfite reductase CysI, the NADP-dependent malic enzyme MaeB, a high-affinity glucose transporter, the nitrilase Nit4 responsible for 3-cyanoalanine assimilation, and the putative regulatory *mocR* gene [13,22]. The genome of *P. pseudoalcaligenes* CECT 5344 also contains genes for two additional nitrilases, and the *cynFABDS* gene cluster involved in the metabolism of the cyano-derivative cyanate [23]. In recent years, bacterial genomes deposited in NCBI have been reannotated using the Prokaryotic Genome Annotation Pipeline (PGAP), and the Average Nucleotide Identity (ANI) values have been calculated to confirm or challenge the initially assigned genus and species [24]. During this process, the *P. pseudoalcaligenes* species has been reclassified as a later heterotypic synonym of *P. oleovorans*. Therefore, the cyanotrophic strain CECT 5344, which was initially identified as a strain of *P. pseudoacaligenes* according to its 16S rRNA [25], has been renamed as *P. oleovorans* CECT 5344 in the NCBI database.

In this work, a comparative genomic analysis has been applied to update the taxonomy of the current *P. oleovorans* species and to establish the position of the cyanotrophic strain CECT 5344 within this species. Furthermore, the pan-genome of the species *P. oleovorans* was achieved, allowing the study of the gene distribution for those involved in cyanide or cyanate assimilation and cyanide resistance in this species and in the genus *Pseudomonas*. Bioinformatic data and further analysis presented in this work constitute a rapid and powerful tool to predict a cyanotrophic phenotype in bacteria, and for the identification of novel genes with a putative role in cyanide biodegradation.

## 2. Results

### 2.1. General Features of Pseudomonas oleovorans Genomes

Out of the 36 genomes currently included in the *P. oleovorans* species (NCBI Dataset), 27 of them were selected after discarding unannotated, redundant, or atypical genomes. When these 27 genomes were loaded into the PanACoTA software (v1.3.1), they were subjected to quality controls about sequencing, assembly, and taxonomy, selecting finally 21 genomes. Although the criteria of exclusion affected the number of genomes analyzed, they contributed to the accuracy, reliability, and consistency of this study. After considering an average nucleotide identity (ANI) cutoff of 96% for all genomes of the same species [26,27], 18 out of 21 selected genomes (85.7%) seemed to be correctly assigned to the *P. oleovorans* species, displaying ANI values > 97% (Figure 1). However, inconsistencies were observed in the genomes of strains T113, E1205, and AG1002, with ANI values of 91%, 87%, and 80%, respectively. In total, five groups of *P. oleovorans* genomes were distinguished in the ANI analysis, and, in this analysis, the genome of *P. pseudoalcaligenes* CECT 5344 clustered with the genomes of GD04039, GD03646, and PO_271 strains (Figure 1). Overall, the size of the 18 *P. oleovorans* genomes ranged from 4.4 to 5.0 Mb (4.6 Mb average), with GC contents from 61.5 to 63.0% (62.0% average) and including 4516 genes and 4266 protein-coding genes as averages (Appendix A).

### 2.2. Phylogenomics of P. oleovorans Species

In some bacterial groups, such as the *Pseudomonas* genus, the resolution of the 16S rRNA gene sequence is insufficient to discriminate closely related species [28]. In this work, an in silico design for taxon delineation at the subspecific level based on digital DNA:DNA hybridization (dDDH) was used. This allowed us to establish evolutionary relationships at the genomic level through the Genome BLAST Distance Phylogeny method (GBDP). In this analysis, the 18 *P. oleovorans* genomes, and the three genomes incorrectly assigned to this species (T113, E1205, and AG1002), were submitted to the Type (Strain) Genome Server (TYGS). The pairwise dDDH estimates of these 21 genomes, and the 31 type strains which were automatically included as the closest neighbors, are listed in Appendix A. The phylogenetic tree inferred from the pseudo-bootstrapped intergenomic distances is shown in Figure 2. The GBDP tree showed a 62.8% average branch support and a δ statistic value of 0.159, which implies a high phylogenetic accuracy. The 18 *P. oleovorans* strains were included in a species cluster, yielding dDDH values ≥ 70%, with the type-strain *Pseudomonas pseudoalcaligenes* NBRC 14167 (Appendix A). The genomes were organized in nine subspecies clusters, one of which included exclusively the cyanotrophic strain CECT 5344. This strain was surrounded by subclusters containing DSM 28630 and GD04000 strains, as well as GD03705, GD03704, and GD04132 strains (Figure 2). The T113 strain showed a maximum dDDH value slightly below 70% (68.1%) when hybridized with the type-strain *P. sediminis* P111, while it yielded dDDH values ≤ 60% with the *P. oleovorans* genomes (Appendix A). The strain AG1002 yielded 90.8% and 79.5% dDDH values with the type-strains *P. oryzihabitans* DSM 6835 and *P. psychrotolerans* DSM 15758, respectively (Appendix A). The dDDH values of the strain AG1002 with the rest of *P. oleovorans* genomes were <17% (Appendix A). In the case of the strain E1205, a maximum dDDH value of 66.6% with the type-strain *P. composti* CCUG 59231 was obtained (Appendix A).

### 2.3. Determination and Characterization of the P. oleovorans Pan-Genome

Of the 36 genomes included in the *P. oleovorans* species in the NCBI Datasets, only 18 genomes met both the taxonomic requirement of belonging to the *P. oleovorans* species (ANI values > 97%) and the sequence and annotation quality filters. Therefore, these 18 genomes were used for the generation of the *P. oleovorans* pan-genome. From a total of 9012 gene clusters identified, 2736 (30.4%) were defined as the core genome (genes present in all genomes). From the rarefaction curve showing the number of protein clusters that are subsequently discovered as more genomes are added to the dataset, an α value (Heap’s Law) of 0.73 was calculated (Appendix A). Dispensable genes (present in several but not all genomes) and singletons (strain-specific genes) constitute the accessory genome. In this analysis, 4191 clusters (46.5%) were considered dispensable, while 2085 (23.1%) were strain-specific clusters. These results were obtained considering a 70% sequence identity in BLAST pairwise alignments. The relationships established among the *P. oleovorans* strains according to the sharing pattern of dispensable clusters (Figure 3) corroborated the previously described phylogenetic study based on the ANI analysis.

Among the dispensable clusters, up to 556 were shared by the strains NCTC10692, RS1, DSM 1045, and NBRC 13583 (Appendix A). The strains NCTC10860 and NBRC 14167 also shared a large number of clusters, accounting for 407 (Appendix A). The cyanotrophic strain *P. pseudoalcaligenes* CECT 5344, onwards to *P. oleovorans* CECT 5344, shared 98 clusters with the strains NCTC10860, RS1, and NCTC 10692, and other 72 genes with all *P. oleovorans* strains except DSM 28630 (Appendix A). The distribution of strain-specific clusters showed that strains PO 271, CECT 5344, GD03646, GD04039, and NCTC10860 contained the highest number of singletons, 468, 368, 295, 266, and 153, respectively, while these genes were particularly scarce in the genomes of the strains GD04132 (7), GD03705 (13), and DSM 1045 (17) (Appendix A). The rest of the strains showed between 35–78 strain-specific clusters (Appendix A). The specific genes of the strain CECT 5344 were distributed within the genome (Figure 4), but four regions were concentrated with 15 (BN5_0137-0152), 18 (BN5_1685-1703), 36 (BN5_1893-1930), and up to 73 (BN5_3803-3881) specific genes (Appendix A).

A gene set enrichment analysis (GSA) contributes to the identification of relevant biological processes or pathways from a set of genes obtained in genome-wide studies. In this work, a GSA of 1208 dispensable genes and 368 strain-specific genes from *P. oleovorans* CECT 5344 was performed using Cluster of Orthologous Groups of proteins (COG), Gene Ontology (GO), and KEGG annotations. The analysis of the dispensable genes of the strain CECT 5344 showed enrichment in the COG functions “replication, recombination and repair” (L) and “secondary metabolites biosynthesis, transport, and catabolism” (Q). More specifically, the enriched biological functions corresponded to the DNA metabolic process/recombination, DNA integration, ncRNA metabolic process, RNA-directed DNA polymerase activity, cellular biosynthetic process, organic nitrogenous compounds biosynthetic process, nitrate and carboxylic acid metabolic processes, methyltransferase activity, and response to mercury ion/transmembrane transport (Figure 5; Appendix A). Focusing on the KEGG metabolic pathways, the dispensable genes of the CECT 5344 strain were enriched in two-component systems (up to 49 genes were identified in this category), biofilm formation, exopolysaccharide and polyketide sugar unit biosynthesis, and xylene degradation (Appendix A).

Regarding the specific genes of the strain CECT 5344, the COG categories enriched corresponded to “transcription” (K) and “defense mechanisms” (V). The enrichment analysis based on GO terms included several biological functions related to DNA, like DNA modification, DNA integration, the DNA metabolic process, and the nucleic acid metabolic process; defense mechanisms, like type I site-specific deoxyribonuclease activity; and others, such as cysteine synthase activity and oxidoreductase activity acting on paired donors with incorporation or reduction of molecular oxygen, with NAD(P)H as one donor, and the incorporation of two atoms of oxygen into one donor (Figure 5). In addition, the specific genes of the CECT 5344 strain were enriched in the KEGG pathways “nitrogen metabolism”, “cysteine and methionine metabolism”, “phenylalanine metabolism”, and “xylene degradation” (Appendix A).

### 2.4. Importance of Hydrolytic Pathways for Cyanide Biodegradation in Bacteria

Hydrolytic pathways are found among the mechanisms that bacteria use to assimilate cyanide [15,16,17]. In the *Pseudomonas* genus, the strain *P. stutzeri* AK61 has been described to degrade cyanide through the cyanide dihydratase CynD [29] (1), while the nitrilase NitC, which catalyzes the hydrolysis of organic cyanides or nitriles (2), is the responsible for cyanide assimilation in *P. oleovorans* CECT 5344 and *P. fluorescens* NCIMB 11764 [18,19]. The nitrile acting as a substrate of NitC is formed by the chemical reaction between cyanide and oxoacids [18].
HCN + 2 H_2_O → HCOOH + NH_3_(1)
RCN + 2 H_2_O → RCOOH + NH_3_(2)

To find out which of these systems is the most widespread among the genus *Pseudomonas* and other bacteria, a BLASTP analysis of sequence alignments was carried out using the CynD or the NitC amino acid sequences from the strains AK61 or CECT 5344, respectively. The database used was the nr_clustered data one that groups amino acid sequences (within 90% identity and 90% length). Minimal thresholds of 50% for sequence identity and 80% for sequence coverage were established. In this analysis, 51 protein clusters containing 301 amino acid sequences homologous to CynD were identified (Appendix A). A small fraction of these homologs belonged to the phylum Pseudomonadota, including the *Alcaligenes*, *Burkholderia*, *Comamonas,* and *Rhodopseudomonas* genera, but a great majority was found in the phylum Bacillota (Firmicutes). The highest number of homologs were identified in *Bacillus* species like *B. safensis* (62 homologs), *B. pumilus* (35), *B. thuringiensis* (16), and *B. cereus* (10), in the family *Paenibacillaceae* (23), and in the class Clostridia (10) (Appendix A). In the case of NitC homologous proteins, 994 protein clusters comprising 3360 amino acid sequences were identified (Appendix A). The NitC homologs were distributed through eight taxonomic groups, including Pseudomonadota, Myxococcota, PVC group, Acidobacteriota, Terrabacteria group, Bacteroidota, Nitrospiraceae, and Desulfobacterales, but the majority belonged to the phylum Pseudomonadota (Appendix A). Unlike CynD, NitC homologs were not found in the phylum Firmicutes. In the genus *Pseudomonas*, NitC homologs were extended among 16 species; meanwhile, the CynD protein was only identified in *P. stutzeri* AK61. Interestingly, the CynD from the strain AK61 presented its maximum identity (99%) with homologs found in beta-proteobacteria (*Alcaligenes* genus; WP_250756510.1). Other homologs found were in the Gram-positive species *Virgibacillus proomii* (WP_077320578.1) and *Brevibacillus laterosporus* (WP_113757746.1), which presented 82% and 80% identity, respectively.

### 2.5. Distribution of Genes Involved in Cyanide Resistance and Cyanide or Cyanate Assimilation in the Pseudomonas Genus

Among the cyanotrophic strains described up to now, only the strain CECT 5344 has been genetically characterized for the resistance to cyanide, which has been attributed to the cyanide-insensitive oxidase encoded by *cioA3/cioB3* genes (BN5_1902-BN5_1903) [13]. In the CECT 5344 and NCIMB 11764 strains, it was experimentally demonstrated that the nitrilase NitC is essential for the assimilation of cyanide [18,19]. Furthermore, these two strains have been described to assimilate the cyano-derivative cyanate through the cyanase CynS [30,31]. The genes coding for CioAB, NitC and CynS were searched in the pan-genome of *P. oleovorans*, as well as in the pan-genomes generated with the *P. fluorescens* (Appendix A) and *P. monteilii* (Appendix A) species. A pan-genome was also generated at the level of the *Pseudomonas* genus (Appendix A), for which genomes of the three cyanotrophic strains and the reference genomes of the genus were used. Appendix A shows the results obtained for all pan-genomes generated in this work.

#### 2.5.1. Cyanide Resistance Genes

In addition to the *cioA3*/*cioB3* genes that code for a functional cyanide-insensitive terminal oxidase required for bacterial respiration in the presence of cyanide, the genome of *P. oleovorans* CECT 5344 harbors the *cioA1*/*cioB1* (BN5_0627-BN5_0628) and *cioA5*/*cioB5* (BN5_2522-BN5_2523) genes, which code for two putative cyanide-insensitive oxidases [32]. In the pan-genome of *P. oleovorans* species, the *cioA3*/*cioB3* genes were identified among the specific genes of the strain CECT 5344 (Figure 6, Appendix A), in a fragment containing 36 singletons which included a cluster composed of 14 genes, three upstream and nine downstream of *cioA3*/*cioB3* [32]. Homologous genes to *cioA3/cioB3* were also identified in the cyanotrophic strain *P. monteilii* BCN3 and in *P. kermanshahensis* Mr36, which does not contain the *nit1C* genes (Figure 6). However, in these two strains, the *cioA3/cioB3* genes were surrounded by different genes when compared to the CECT 5344 strain.

Concerning the other cyanide-insensitive oxidases encoded within the genome of the strain CECT 5344, the *cioA1/cioB1* genes were part of the core genome of *P. oleovorans* and they were present extensively in the rest of the genomes analyzed, while the *cioA5/cioB5* genes occurred with an intermediate frequency (Figure 6). The alignment of amino acid sequences showed that CioA1/CioB1 exhibited a higher identity to CioA3/CioB3 proteins (67/65%) than to CioA5/CioB5 (30/27%). In *P. oleovorans* CECT 5344, the cyanide assimilation and resistance mechanisms have been described to be linked through a malate:quinone oxidoreductase, which reduces malate to oxaloacetate with the electrons received from CioA3/CioB3 [14]. The genome of the strain CECT 5344 harbors two genes (BN5_0860 and BN5_1358) that code for the malate:quinone oxidoreductases MqoA and MqoB, respectively. Both genes belonged to the core-genome of *P. oleovorans* species, while only *mqoA* was mostly identified in the rest of the *Pseudomonas* genomes analyzed (Figure 6).

#### 2.5.2. Assimilation of Cyanide

The *nitC* gene belonged to the dispensable genes of the strain CECT 5344 and it was identified in seven genomes (39%) of the *P. oleovorans* species, including CECT 5344, GD03646, GD04039, NBRC 13583, DSM 1045, NCTC10692, and RS1 (Figure 6). In these strains, the *nitC* gene was included in the *nit1C* gene cluster, which presented the same gene organization as the strain CECT 5344 (Appendix A). The sequence alignments of Nit proteins showed 100% identity when compared to their homologs of *P. oleovorans* strains (Appendix A).

Considering the pan-genome of the other species in which cyanotrophic strains have been described, the frequency of the *nitC* gene was lower than in the *P. oleovorans* species (*P. fluorescens* 6% and *P. monteilii* 5%). The *nitC* gene was present specifically in the *P. fluorescens* NCIMB 11764 and 2P24 strains and, also, in *P. monteilii* BCN3. In these pan-genomes, *nitA*, *nitB*, *nitD,* and *nitH* gene homologs were found, but homologs to *nitE*, *nitF,* and *nitG* genes were not identified. However, a manual search allowed the identification of homologs to these three last genes, although showing amino acid identities lower than 70% when compared to the corresponding genes of the strain CECT 5344 (Appendix A). NitB homologs showed the highest amino acid sequence identity (>90%) (Appendix A). The *nit1C* gene clusters of the NCIMB 11764, 2P24, and BCN3 strains showed the same gene arrangement as the present in the genomes of the *P. oleovorans nit1C*-containing strains, except for the NCIMB 11764 strain, in which the regulatory gene *nitA* was located outside the *nit1C* gene cluster (three genes downstream *nitH*), and orientated in the same direction as the structural *nit* genes (Appendix A). In the pan-genome of the reference genomes of the *Pseudomonas* genus, nine genomes containing *nitC* homologs were identified. Four of them were located in the *nit1C* cluster (*P. mandelii* KGI MA19, *P. arsenicoxydans* CECT 7543, *P. reinekei* BS3776, and *P. viridiflava* CFBP 1590), and showed the same *nit1C* gene cluster organization as in the CECT 5344 strain (Appendix A). The sequence identity pattern of the Nit proteins compared to those of *P. oleovorans* CECT 5344 was similar to that described for the strains NCIMB 11764, 2P24, and BCN3 (Appendix A).

To establish the total distribution of the *nit1C* gene cluster in the *Pseudomonas* genus, a restricted BLASTP search was performed using, as the protein query, the nitrilase NitC from the strain CECT 5344 against the database “Refseq Select proteins”. The results showed 31 sequences sharing identity values within the range 97.85–70.59% (with a minimum sequence coverage of 94%), while the rest of the sequences showed a much lower percentage of identity (49–32%) (Appendix A). A phylogenetic analysis of the 31 sequences with the highest identity revealed that these homologs were distributed in two different clades. One clade contained 21 sequences, including the NitC sequence of the strain CECT 5344, with identity values 100–83%. The second clade comprised 11 sequences with identity values 78–70.59% (Appendix A). The genetic context of the genes that code for the proteins included in the first (largest) clade revealed that these genes were part of the *nit1C* gene cluster. However, the *nitC* genes that clustered in the second clade presented a different genetic context, which included, in all cases, the adjacent *glxA* gene coding for a transcriptional regulator, located upstream from *nitC* and displaying the same orientation. The nitrilase sequences encoded by the *nit1C* gene clusters belonged to 25 identified *Pseudomonas* species (Table 1), as well as to several non-identified species. *P. viridiflava*, *P. oleovorans,* and *P. reinekei* were the species with the largest number of strains containing the *nit1C* gene cluster, with 11, 7, and 4, respectively (Table 1). However, when the number of genomes containing the *nit1C* cluster was considered relative to the number of sequenced genomes in each species, the importance of *P. viridiflava*, *P. oleovorans,* and *P. reinekei* decreased considerably (Table 1).

In addition to the nitrilase NitC that is essential for cyanide assimilation, the genome of *P. oleovorans* CECT 5344 harbors three other nitrilase genes, including *nit4* (BN5_1912), *nit1* (BN5_1925), and *nit3* (BN5_4427). The nitrilase Nit4, encoded by the *cioAB-nit4* gene cluster, is involved in the assimilation of the cyano-derivative 3-cyanoalanine [22], while the other two nitrilases have not been functionally characterized. In the pan-genome of *P. oleovorans*, genes that code for these three nitrilases were included among the singletons of the strain CECT 5344 (Figure 6). In the rest of the genomes analyzed, neither *nit1* nor *nit4* homologs were found, while *nit3* homologs were identified in *P. citronellolis* P3B5, *P. paraeruginosa* Cr1, *P. alkylphenolica* Neo, *P. multiresinivorans* populi, *P. nitroreducens* L4, and *P. brenneri* BS2771 (Figure 6).

#### 2.5.3. Assimilation of Cyanate

*P. oleovorans* CECT 5344 has also been described to assimilate the cyano-derivative cyanate, a less toxic compound than cyanide [31]. Cyanate is assimilated through the cyanase CynS, which catalyzes the bicarbonate-dependent cyanate hydrolysis producing ammonium and carbon dioxide. In the strain CECT 5344, the cyanase CynS and the ABC-type cyanate transporter CynABD are encoded by the *cyn* gene cluster, which is under the control of the product of the *cynF* gene [23]. Initially, the *cynS* gene was included in the strain-specific genes of *P. oleovorans* CECT 5344, even when the analysis included the *P. fluorescens* NCIMB 11764 strain, in which the cyanase had been described [30]. A BLAST analysis, performed using the amino acid sequence of CynS from the strain CECT 5344 as the query sequence, revealed the existence of three *cynS* homologs in the genome of *P. fluorescens* NCIMB 11764, with identity values of 47.18%, 44.06%, and 43.66%. On the basis of this BLAST analysis, homologs to *cynS* with a low amino acid sequence identity (<60%) were identified in the *nit1C*-containing genomes of *P. oleovorans* GD04039, *P. fluorescens* 2P24, and *P. monteilii* BCN3, and, also, in the genomes of strains lacking the *nit1C* cluster like the *P. oleovorans* GD04132, GD03704, and GD03705 strains, *P. silesiensis* A3, and *P. izuensis* LAB_08 (Figure 6). These clusters containing *cynS* genes presented a different gene context from that of *P. oleovorans* CECT 5344, with most of them containing a gene coding for the carbonic anhydrase. In the pan-genome constructed with the reference genomes of the *Pseudomonas* genus, *cynS* was identified as a dispensable gene in *P. sihuiensis* KCTC 32246 (98.63% identity), *P. toyotomiensis* SM2 (97.26%), *P. khazarica* ODT_83 (93.84%), *P. sediminis* B10D7D (90.41%), and *P. campi* S1-A32-2 (79.45%). When the amino acid sequence identities were higher than 90%, compared to *P. oleovorans* CECT 5344, the genetic context of the *cynS* genes was similar to that of this strain. In *P. sediminis* B10D7D, the *cynS* gene was not included in a gene cluster.

#### 2.5.4. Searching for Novel Genes Involved in the Biodegradation of Cyanide

Considering that the *nit1C* gene cluster has been described as essential for cyanide assimilation, a co-occurrence analysis with this gene cluster was performed to identify putative genes involved in this biodegradative process. The *nit1C*-containing genomes were selected at the level of the species or genus to retrieve core-specific genes of this subset. In the pan-genome of *P. oleovorans*, 10 genes were specifically present in all the *nit1C*-containing genomes, and they were absent in the non-*nit1C* genomes. From these genes, eight belonged to the *nit1C* gene cluster and the other two, coding for a methyl-accepting chemotaxis sensory transducer (BN5_3010) and a disulfide isomerase-like protein (BN5_3011), were located next to each other. Considering a lower positive stringency (genes present specifically in six out of seven *nit1C*-containing genomes), the BN5_4084, BN5_4086, and BN5_4088 genes that encode proteins of unknown function were identified. When the lowest stringency considered was negative (genes linked to all *nit1C* genomes but present in up to two non-containing *nit1C* genomes), the BN5_3887 gene, which codes for a hypothetical protein, was identified. When the co-occurrence analysis was performed in the species *P. fluorescens*, two genes were also identified, in addition to those encoded by the *nit1C* cluster. These two genes were located at two close *loci* (B723_18675 and B723_18680), and they code for a protein of unknown function and a protein putatively involved in signal transduction, respectively. In the *P. monteilii* species, only the BCN3 strain contained the *nit1C* gene cluster, and, therefore, a co-occurrence analysis was not possible. Additionally, genes were not found in a co-occurrence analysis performed with the genomes of the *P. oleovorans*, *P. fluorescens,* and *P. monteilii* species, either when only the three cyanotrophic strains were selected or when all the genomes containing the *nit1C* gene cluster were included.

## 3. Discussion

### 3.1. Genomics Supports Taxonomy Position of the Cyanotrophic Strain CECT 5344 as a Pseudomonas oleovorans Species

In the last years, the application of comparative genomics to phylogenetic analysis is allowing a more accurate taxonomic classification of bacteria. The taxon *Pseudomonas* is one of the genera that has experimented with more changes regarding taxonomy. In a recent review of the boundaries of the *Pseudomonas* genus, a novel genus *Stutzerimonas* and the transfer of species to the genus *Halopseudomonas* that belongs to the *Pseudomonas pertucinogena* group have been proposed based on previous phylogenetic studies and considering that genera have to be monophyletic and species within genera have to share a high ratio of orthologous genes calculated by indices, such as the Jaccard index [33]. The genus *Stutzerimonas*, comprising strains in the formerly phylogenetic group *Pseudomonas stutzeri*, has been described through a detailed phylogenomic analysis [34]. *P. pseudoalcaligenes* is a species that has been exposed to different reorganizations since it was first described [35]. Although *P. pseudoalcaligenes* was initially considered a close but different species to *P. oleovorans*, it is currently considered as a synonym of *Pseudomonas oleovorans*. In the intrageneric structure of the *Pseudomonas* genus previously resolved [36], considering the DNA gyrase-B subunit *gyrB* and sigma70 factor *rpoD* genes, the *P. pseudoalcaligenes* and *P. oleovorans* species were included in one of the clusters of the “*P. aeruginosa* complex”. In a more accurate analysis, using a multilocus sequence analysis (MLSA), regarding 16S rRNA, *gyrB*, *rpoB*, and *rpoD* genes, a *P. oleovorans* group that includes the species *P. oleovorans*, *P. pseudoalcaligenes*, *P. alcaliphila,* and *P. mendocina* was proposed [37]. However, based on DNA-DNA relatedness and phenotypic and biochemical data, a reclassification of the strain *P. pseudoalcaligenes* ATCC 17440 as a synonym of *P. oleovorans* ATCC 8062 was proposed [38]. The cyanotrophic strain CECT 5344 was identified as a member of the species *Pseudomonas pseudoalcaligenes* based on its 16S rRNA sequence [25]. Later, according to a four-gene MLSA, it was shown that this strain has a 99.81% similarity with the *P. oleovorans* subsp. *oleovorans* ATCC 8062 type strain [39]. Recently, the application of more robust taxonomic methods as the digital whole-genome comparisons by using Average Nucleotide Identities (ANIs) has confirmed this reclassification, and the strain CECT 5344 has been renamed as *P. oleovorans* CECT 5344. In this work, different phylogenomics approaches have been performed to update the taxonomy of the *P. oleovorans* species and to establish the position of the strain CECT 5344. Out of the 21 genomes assigned to the *P. oleovorans* species, the T113, E1205, and AG1002 strains showed ANI values below the species cutoff (96%), and dDDH estimations with the genome of the type-strain *P. pseudoalcaligenes* NBRC 14167, also below the species cutoff value (70%) (Figure 1, Appendix A). These results provide evidence of mistakes in the taxonomy of the *P. oleovorans* species, and it is proposed that the strain AG1002 should be reassigned to the species *Pseudomonas oryzihabitans*, while the strains T113 and E1205 could be considered as potential new species. The strain CECT 5344 presented pairwise ANI values greater than 96% with the remaining 17 genomes of *P. oleovorans* (Figure 1). In addition, in the GBDP phylogenetic study, these 18 genomes were included on a species cluster together with the type-strain *P. pseudoalcaligenes* NBRC 14167 (Figure 2). In this analysis, the 18 strains were organized in nine subspecies clusters, which agree with the clusters obtained in the dendrograms resulting from the ANI analysis and from the sharing pattern of the dispensable clusters of *P. oleovorans* species (Figure 1 and Figure 3). The strains belonging to the *P. oleovorans* species were distributed in two main clades, one with the strains NCTC10692, RS1, DSM1045, and NBRC 13583, and the other with the remaining 14 strains. The CECT 5344 strain was located close to the strains GD04039, GD03646, and PO_271 (Figure 2), although they do not form a subspecies cluster. These results confirm the reclassification of the strain CECT 5344 into the species *P. oleovorans* and clarify the boundaries and the taxonomic structure of this species. In addition, the results obtained in this work using ANI, dDDH, and the sharing pattern of the dispensable clusters were highly concordant, suggesting that this is a valid approach for determining the taxonomic structure of the species *P. oleovorans*.

### 3.2. Defining the Pan-Genome of Pseudomonas oleovorans

Members of the *Pseudomonas* genus have been the subject of multiple comparative genomic studies with the aim of understanding the genetic basis for diversity and adaptation to different environments. These studies were focused primarily on pathogens and species of interest to be applied in environmental biotechnology approaches. Thus, pan-genomes have been described for the human pathogen *P. aeruginosa*, the phytophatogenic *P. syringae*, the plant growth-promoting rhizobacteria *P. fluorescens*, and the species effective at removing pollutants *P. putida* [40]. In this work, the pan-genome of *P. oleovorans*, a species that includes the relevant cyanotrophic strain CECT 5344, has been defined for the first time. Using a filtered set of genomes (*n* = 18, NCBI Datasets, 10 December 2023), the *P. oleovorans* pan-genome was determined, obtaining a total of 9012 clusters, from which 2736 were considered part of the core genome. Genomes are considered open if the pan-genome size increases as more genomes are included in the analysis, due to the contribution of novel genes by new strains. Heap Low determines an α value below 1 for open genomes [41]. In the case of *P. oleovorans*, the size of its pan-genome increased with the number of genomes added to this analysis (Appendix A), yielding the calculated α value of 0.73, thus indicating the open property for this species. The low percentage of the core genome of *P. oleovorans* (30.4%) with respect to the entire pan-genome is consistent with open genomes, such as that described for *P. parafulva* (17%) or *P. putida* (54%) [42,43]. The open pan-genome of *P. oleovorans* indicates that this species is characterized by a high genomic heterogeneity among its strains. Furthermore, open pan-genomes are characteristic of species with a free-living lifestyle and high metabolic flexibility, allowing the colonization of a variety of different ecological niches. The strains whose genomes have been analyzed were described to be isolated from diverse sources, including sludge from rivers, sink in hospitals, industrial wastewaters, and even the mouse gut. Metabolic flexibility confers to bacteria the ability to adapt to different environmental conditions and to use a broad range of molecules as nutrients, even pollutants or toxic compounds. The strain DSM 1045 has been described to use hydrocarbons [44], and the strain CECT 5344 is characterized by its capacity to assimilate cyanide and detoxify arsenite and heavy metals [15,16,17,25].

Gene sharing is determined mainly by ecological aspects, such as the co-occurrence and diversity of species present in a habitat [45]. In addition, the expanding capacity of a pan-genome can also be influenced by intrinsic barriers to horizontal gene transfer, such as the CRISPR system [46]. A recent analysis in the genus *Pseudomonas* showed that the CRISPR systems are only present in 17.5% of the analyzed genomes, accounting for up to 19.6% of all the *Pseudomonas* species [47]. In *P. aeruginosa*, around 50% of sequenced genomes contain CRISPR-Cas systems (subtypes I-F, I-E, and I-C), which have been demonstrated to restrict the acquisition of mobile genetic elements [48]. In the core genome of *P. oleovorans*, genes coding for a class 1 subtype I-F CRISPR-Cas system were identified (Appendix A). In addition, many genes coding for additional subtypes I-E, I-C, and others were included in the accessory genome of *P. oleovorans* (Appendix A). Although the presence of CRISPR-Cas systems could suggest a limitation for the evolution of the pan-genome of *P. oleovorans*, transcriptomic studies of the CECT 5344 strain revealed that these systems were repressed under cyanotrophic conditions [49]. Thus, the acquisition of exogenous DNA under unfavorable conditions could contribute to bacterial survival, and, hence, cyanide could be proposed as a key factor driving the genetic diversity and contributing to the expansion of the pan-genome of *P. oleovorans*.

### 3.3. The Nitrilase NitC Essential for the Cyanotrophic Phenotype Is Broadly Extended in Bacteria and It Is Encoded in the Accessory Genome of the Pseudomonas Genus

In the *Pseudomonas* genus, the ability to assimilate cyanide was first described in the strain AK61 of *Pseudomonas stutzeri*, in which the cyanide dihydratase CynD catalyzes the hydrolysis of cyanide to ammonia and formate [29,50]. This enzyme showed a high sequence identity with members of the phylum Firmicutes, in which CynD is mainly restricted (Appendix A). However, the maximum sequence identity to CynD from the strain AK61 was with members of the *Alcaligenes* genus, included in the phylum beta-proteobacteria (Appendix A). These results, together with the absence of cyanide dihydratases homologs in the *Pseudomonas* genus, suggest that the presence of CynD in the strain AK61 was probably the result of a punctual event of the gene transfer from a member of the Firmicutes phylum. Other strains of *Pseudomonas* that were described to show a cyanotrophic phenotype were *P. pseudoalcaligenes* CECT 5344, *P. fluorescens* NCIMB 11764, and *P. monteilii* BCN3 [18,19,51]. The CECT 5344 and NCIMB 11764 strains have been demonstrated to possess a hydrolytic pathway, in which a cyanohydrin is hydrolyzed into ammonium by the nitrilase NitC, which is encoded by the conserved *nit1C* gene cluster. In the strain BCN3, cyanotrophy was also attributed to the presence of the *nit1C* gene cluster. By the time the *nit1C* gene cluster from the strain CECT 5344 was described to be required for cyanide biodegradation [18], homologs to the genes present in this cluster were not identified in other Pseudomonads. Currently, 49 genomes from 25 of the *Pseudomonas* species harbor the *nit1C* gene cluster, with a genetic structure mostly consistent with that of *P. oleovorans* CECT 5344 (Table 1). Considering that the *nit1C* gene cluster confers a cyanotrophic phenotype, its presence in genomes reflects the probable significance of these genes in the adaptation to ecological niches containing cyanide. In recent metagenomic studies focused on marine picocyanobacteria, the *nit1C* gene cluster has been found to be differentially distributed both throughout the water column and oceanwide. Thus, this gene cluster was more abundant in the surface of the ocean, where cyanide is produced by cyanogenic micro-organisms, as well as in warm, iron-replete waters, particularly in low-N areas like the Indian ocean [21,52]. This differential distribution of the *nit1C* gene cluster suggests a niche-related adaptive strategy. Among the *Pseudomonas* species with a large number of genomes containing the *nit1C* gene cluster (Table 1), *P. viridiflava* is a phytopatogenic species that colonize plants, organisms where cyanogenesis is broadly extended. On the other hand, some strains of *P. reinekei* have been described to produce cyanide [53]. In *P. oleovorans,* the *nit1C* gene cluster is encoded by the accessory genome of seven out of the 18 genomes analyzed. The cyanotrophic strain CECT 5344 was isolated from the Guadalquivir River (Córdoba, Spain), where the jewelry industry dumped cyanide-containing wastewaters [25]. The *P. oleovorans* strains GD03646, GD04039, NCTC10692, RS1, DSM 1045, and NBRC 13583 can be predicted to have a cyanotrophic phenotype as a consequence of the acquisition of the *nit1C* gene cluster in cyanide-containing environments, where the nitrilase NitC could confer an adaptive advantage by detoxifying cyanide and releasing ammonium that can be incorporated to organic nitrogen to support cell growth. Hypothetically, the presence of cyanide could influence the evolution of the genomes. However, the two clusters into which the *nit1C* genomes were grouped based on the presence/absence profile of the accessory genome may reflect another important factor in genome evolution (Figure 3).

### 3.4. Genetic Factors Conferring Cyanide Resistance Are More Variable and Extended than Those for Cyanide Assimilation in the Pseudomonas Genus

Although the assimilatory pathway contributes to cyanide detoxification, a cyanotrophic phenotype requires other mechanisms that confer resistance to cyanide, such as the cyanide-insensitive terminal oxidase. Among cyanotrophic bacteria, the resistance to cyanide has only been studied in the CECT 5344 strain, in which this ability was attributed to the *cioA3/cioB3* genes [13], one of the three paralogs of the cyanide-insensitive oxidase identified in the genome of this strain [32]. In *P. oleovorans*, the *cioA3/cioB3* genes were included in the accessory genome (Figure 6, Appendix A). Although the strains BCN3 and Mr36 showed *cioA3/cioB3* homologs, these genes were included in a different genetic context to that shown by the strain CECT 5344. The *cioA3B3-nit4* cluster is part of the specific-genes of the strain CECT 5344, even at the level of the *Pseudomonas* genus (Figure 6). This cluster also includes the *nit4* gene that codes for a nitrilase that uses 3-cyanoalanine as a substrate [22]. Homologs to this gene were absent in the cyanotrophic strains NCIMB 11764 and BCN3, and in the rest of genomes analyzed (Figure 6), and, therefore, it can be deduced that these strains do not display the ability to assimilate the cyano-derivative 3-cyanoalanine. In a more extensive analysis, we confirmed that there is not a gene cluster homologous to the *cioA3B3-nit4* genes of the strain CECT 5344 in the bacterial databases; thus, it can be assumed that this cluster was punctually transferred to the strain CECT 5344 from a bacterial strain that has not been identified and/or sequenced up to now.

In addition to the *cioA3/cioB3* homologs identified in the strain BCN3, the presence of *cioA1/cioB1* in the BCN3 and NCIMB 11764 strains (Figure 6) suggests that the cyanide-insensitive terminal oxidase CioA1/CioB1 could be the responsible for cyanide resistance in these cyanotrophic strains. This cyanide resistance system was present in the core genome of the species *P. oleovorans*, *P. fluorescens*, and *P. monteilii* (Appendix A), as well as in the core genome established for the pan-genome of the *Pseudomonas* genus (Appendix A), indicating that the resistance to cyanide is a widely extended phenotype in this genus. In addition, this result denotes that the capacity to assimilate cyanide is always linked to cyanide resistance, but the ability to resist cyanide can be found as a phenotype independent of cyanide assimilation.

In some bacteria, the malate dehydrogenase is replaced by a malate:quinone oxidoreductase (Mqo), which is essential for the tricarboxylic/glyoxylic acid cycles. In *P. oleovorans* CECT 5344, a Mqo that connects cyanide resistance to cyanide assimilation has been described [14]. In this cyanotrophic strain, a malate oxidase system comprising a Mqo activity coupled to the terminal oxidase (with oxygen as the terminal electron acceptor) was not induced in cells grown with cyanide. However, the Mqo was probed to be coupled to the cytochrome *c* oxidase in the absence of cyanide and to the cyanide-insensitive oxidase under cyanotrophic conditions [14]. In the last condition, the oxaloacetate generated by the Mqo reacts chemically with cyanide producing the cyanohydrin, which is the substrate of the nitrilase NitC. Thus, a Mqo involved in cyanide resistance can be proposed when Cio is present, while an additional role of Mqo in cyanide assimilation can be suggested when NitC is also present. The genome of the strain CECT 5344 harbors two genes coding for MqoA and MqoB, suggesting that each could have a specific role in the central or cyanide metabolism. Although these two Mqo have not been functionally characterized up to date, the correlation between *mqoA* and *cioA1/cioB1* genes showed in this work suggests that MqoA could be related to cyanide resistance in most of the genomes analyzed. In those strains containing NitC, such as the cyanotrophic strains NCIMB 11764 and BCN3, the MqoA would also be involved in cyanide assimilation. Interestingly, the genomes of some *Pseudomonas*, such as *P. putida* KT2440, harbor even three Mqo paralogs, some of them showing a high identity/similarity with both MqoA and MqoB of *P. oleovorans* CECT 5344. In the KT2440 strain, Mqo redundancy is also associated to *cioAB* genes, but do not correlate with the presence of *nitC* gene. Further studies in strains with multiple *mqo* genes, such as the KT2440 strain, could reveal new functions of Mqo enzymes.

### 3.5. The Presence of Genes Coding for Other Nitrilase or Cyanase Enzymes Is Not Associated to the Assimilation of Cyanide

In the preliminary analysis of the genome of the strain CECT 5344, genes coding for nitrilases NitC and Nit4 involved in cyanide and 3-cyanoalanine assimilation, and two other genes (BN5_1925 and BN5_4427), which code for non-characterized nitrilases, were identified [32]. These last two genes belonged to the specific genes of the CECT 5344 strain in the pan-genome of *P. oleovorans* and they were absent in the genomes of the strains NCIMB 11764 and BCN3 (Figure 6). Although multiple nitrilases could constitute a specific feature of cyanotrophic bacteria like the CECT 5344 strain, this work discarded the association between a cyanotrophic phenotype and the presence of other nitrilases, like Nit1 or Nit3. In fact, mutants in these two additional *nit* genes of the strain CECT 5344 are still able to assimilate cyanide and 3-cyanoalanine, thus discarding a relevant role of these nitrilases in cyanide degradation [22].

Cyanate was initially proposed as a possible intermediate that could be generated by a specific cyanide monooxygenase during cyanide assimilation, although the existence of this enzyme has not been experimentally demonstrated [25]. Subsequently, the cyanate could be assimilated through cyanase activity. In the CECT 5344 strain, cyanide induces the cyanase CynS, but a mutant lacking this enzyme was not affected in its ability to assimilate cyanide [23,31]. However, the generation of low concentrations of cyanate during cyanide assimilation or the existence of an unspecific monooxygenase induced by cyanide could be regarded. Considering the possibility of cyanate production associated to cyanide assimilation via the nitrilase NitC, the co-existence of the *cynS* gene and the *nit1C* gene cluster is expected. In *P. oleovorans*, the *cynS* gene was included in the accessory genome, and its frequency was similar to that shown by the *nit1C* gene cluster (Figure 6, Appendix A). However, the presence of the *cynS* gene among the genomes of *P. oleovorans* containing the *nit1C* gene cluster was minimal, and it was also found in genomes lacking the *nit1C* cluster. The cyanotrophic strains NCIMB 11764 and BCN3 harbored the *cynS* gene in their genomes (Figure 6), but considering all the genomes analyzed, a significant association between the *cynS* gene and the *nit1C* gene cluster was not found. The sequence identity of the cyanase CynS of the strain CECT 5344 displayed an identity lower than 70% when compared to its homologs from other members of *P. oleovorans* or from the strains NCIMB 11764 and BCN3. In these micro-organisms, the genetic context of the *cynS* was different to that displayed by the CECT 5344 strain. Specifically, in *P. fluorescens* NCIMB 11764 and *P. monteilii* BCN3, three *cynS* paralogs sharing a high amino acid sequence identity among them were identified. By contrast, the identity of CynS from *P. oleovorans* CECT 5344 was higher with homologs found in other species, such as *P. campi*, *P. sediminis*, or *P. toyotomiensis* (Figure 6), in whose genomes the genetic context of the *cynS* gene was the same as that described for the strain CECT 5344. These results suggest that, in the *Pseudomonas* genus, the *cyn* cluster has been acquired from different sources, and that, in some cases, duplication events of the *cynS* gene have occurred.

### 3.6. Identification of Novel Genes with a Potential Role in Cyanide Biodegradation

The comparative genome analysis of cyanotrophic bacteria living in cyanide-containing ecological niches provides an opportunity to understand the molecular mechanisms involved in cyanide detoxification and biodegradation, and it is particularly useful in the identification of novel genes participating in cyanide metabolism. Considering that the *nit1C* gene cluster is essential for cyanide assimilation in *Pseudomonas* species, a genetic co-occurrence strategy considering this cluster was followed with the main purpose of identifying novel genes involved in cyanide biodegradation. In this analysis, no genes linked to the *nit1C* gene cluster were found when the core-specific subset was generated with all the *nit1C*-containing genomes identified in the *Pseudomonas* genus. However, some genes that co-exist specifically in the *nit1C* genomes from *P. oleovorans* or *P. fluorescens* were identified. These results indicate that there are no additional genes to *nit1C* participating in cyanide assimilation through the nitrilase NitC in a general way, but only at the species level, suggesting that these genes are linked to species-specific adaptations to environments with cyanide. Interestingly, in *P. oleovorans,* two genes were identified in all the genomes that specifically contain the *nit1C* gene cluster. The fact that these two genes are located next to each other in these genomes confers a greater reliability on the association of these genes with the *nit1C* gene cluster. The involvement of a protein encoded by one of these genes in chemotaxis could be related with a possible role of cyanide as an attractant in the putative cyanotrophic strains of the *P. oleovorans* species. Other two different genes were also identified as being specific to *nit1C* genomes in *P. fluorescens*. As in the case of *P. oleovorans*, these genes were located together in the genomes, conferring a high significance to their identification. Their implication in signal transduction could be related to a mechanism specifically activated by cyanide in this species. The putative role of these genes in cyanide biodegradation will require further experimental studies.

## 4. Materials and Methods

### 4.1. Genomic Datasets

Whole-genome sequences publicly available (October 2023) in the NCBI Datasets for the *P. oleovorans*, *P. fluorescens,* and *P. monteilii* species were downloaded and used for analysis (Appendix A). Out of 36 genomes available for the *P. oleovorans* species, 27 were selected after discarding unannotated, redundant, and atypical genomes. The genome of the cyanotroph *P. pseudoalcaligenes* CECT 5344 was included among these. In the *P. fluorescens* species, 298 genomes were available, of which 197 were annotated and typical genomes. To reduce this number, only genomes with chromosome or complete assembly levels were considered, resulting in 31 genomes, which included the genome of the cyanotroph *P. fluorescens* NCIMB 11764. The NCBI Datasets comprised 61 genomes for the *P. monteilii* species. Filtering for annotated genomes, scaffold, chromosome, or complete assembly level, and excluding atypical genomes, 17 genomes were selected, including the genome of the cyanotroph *P. monteilii* BCN3. In the *Pseudomonas* genus, the NCBI Datasets comprised 31.237 genomes. For the pan-genome analysis, only 145 reference genomes with chromosome or complete assembly level were considered (Appendix A). Reference genomes are characterized by their high quality and their selection conferred a uniform representation of all species of the genus *Pseudomonas*.

### 4.2. Average Nucleotide Identity Analysis

To explore the genetic distance and relatedness for the *P. oleovorans* genome set containing the CECT 5344 strain, an Average Nucleotide Identity (ANI) analysis was performed using the software JSpecies v.1.2.15 with a threshold of 95% as the cutoff for species [27].

### 4.3. Phylogenomic Analysis

For whole-genome phylogenetic analysis of *P. oleovorans* strains, the methodology based on the Genome BLAST Distance Phylogeny method (GBDP) was followed. This methodology was implemented in the Type Strain Genome Server (TYGS) platform (https://tygs.dsmz.de/, accessed on 25 October 2023), obtaining the phylogeny based on the whole-genome sequences [54]. Determination of closest type strain genomes was carried out in two complementary ways. First, all user genomes were compared against all type strain genomes available in the TYGS database via the MASH algorithm, a fast approximation of intergenomic relatedness [55], and the ten type strains with the smallest MASH distances were chosen per user genome. Second, an additional set of ten closely related type strains was determined via the 16S rRNA gene sequences. These were extracted from the user genomes using RNAmmer [56] and each sequence was subsequently BLASTed [57] against the 16S rRNA gene sequence of each of the currently 20118 type strains available in the TYGS database. This was used as a representative to find the best 50 matching type strains (according to the bitscore) for each user genome and to subsequently calculate precise distances using the Genome BLAST Distance Phylogeny approach (GBDP) under the algorithm ‘coverage’ and distance formula d5 [58]. These distances were finally used to determine the 10 closest type strain genomes for each of the user genomes. For the phylogenomic inference, all pairwise comparisons among the set of genomes were conducted using GBDP and accurate intergenomic distances inferred under the algorithm ‘trimming’ and distance formula d5 [58]. One hundred distance replicates were calculated each. Digital DDH values and confidence intervals were calculated using the recommended settings of the GGDC 4.0 [58,59]. The resulting intergenomic distances were used to infer a balanced minimum evolution tree with branch support via FASTME 2.1.6.1 including SPR postprocessing [60]. Branch support was inferred from 100 pseudo-bootstrap replicates each. The trees were rooted at the midpoint [61] and visualized with PhyD3 [62]. The type-based species clustering using a 70% dDDH radius around each of the 31 type strains was carried out as previously described [54]. Subspecies clustering was carried out using a 79% dDDH threshold as previously introduced [63].

### 4.4. Pan-Genome Analysis and Comparative Genomics

The pan-genome was determined with the PanExplorer application using the GenBank identifiers of the selected genomes as input [64]. The workflow included an initial gene clustering and pan-genome analysis performed with the PanACoTA software, which applies a sequence quality control and mash filtering to avoid redundant and misclassified genomes, and uniformly annotates and builds the pan-genomes [65]. A sequence identity level of 70% was applied in BLAST pairwise alignments performed in PanACoTA, and homolog proteins were then clustered by single-linkage (all members of a cluster have at least one other member of the same cluster within distance threshold from itself). To determine the pan-genome at genus level, a threshold of 70% sequence identity was also applied to avoid clustering of proteins with distinct functions, while specific BLAST analyses were performed when ortholog genes relevant for cyanide metabolism were lost in some genomes. The rarefaction curve was computed by micropan R package and the software IQtree v2.3.x was used to infer phylogeny. The GC content was calculated using SkewIT. The physical map of core-genes and strain-specific genes was constructed as a circular genomic representation using the tool Circos [66].

### 4.5. Functional Analysis

Gene set enrichment analysis was performed with the FUNAGE-Pro web server [67] and the ShinyGO application [68]. Clusters of Orthologous Groups (COG) functional categories were attributed using RPSblast against the COG database in the FUNAGE-Pro web server. This program also provided enrichment analysis of molecular function terms of Gene Ontology (GO) KEGG pathways, keywords, eggNOG, and protein domains (Pfam). Enrichment based on the cellular component terms of GO was performed with the ShinyGO application.

### 4.6. Search for Sequence Homologs

Genomic *loci* with specific gene homologs were identified by a BLAST search at NCBI. To specifically search for the *nit1C* gene cluster in the *Pseudomonas* genus, the amino acid sequence corresponding to the nitrilase NitC of the strain *P. pseudoalcaligenes* CECT 5344 was used as input query against the NR database set as *Pseudomonas* genus. Subsequently, the genes adjacent to the *nitC* gene corresponding to the hits with the highest identity were analyzed.

## 5. Conclusions

The cyanotrophic phenotype found in some bacteria constitutes the basis for the implementation of biotechnological processes focused on the decontamination of cyanide-containing industrial wastewaters. Molecular studies about cyanide biodegradation have been mainly focused on the hydrolytic pathways catalyzed by the cyanide dihydratase CynD or by the nitrilase NitC. In this work, it has been established that NitC is widely distributed among a large number of taxonomic groups, including the genus *Pseudomonas*. A comparative genomic analysis focused on this genus has established, for the first time, the characteristics of the pan-genome of the species *P. oleovorans*, in which has been phylogenomically located the model cyanotrophic strain *P. pseudoalcaligenes* CECT 5344. In addition, it has been possible to conclude that the *nit1C* gene cluster that codes for the nitrilase NitC, essential for cyanide assimilation, is part of the accessory genome of the genus *Pseudomonas*, being located in 49 genomes of 25 species of this genus, with most of the genomes belonging to *P. viridiflava*, *P. oleovorans,* or *P. reinekei*. On the other hand, the resistance to cyanide was more widespread than cyanide assimilation, and the presence of additional nitrilases or the cyanase was not associated to the assimilation of cyanide. This study also allowed the identification of genes related to chemotaxis or signal transduction co-occurring with the *nit1C* cluster, suggesting a relevant participation of these processes during cyanide biodegradation. Overall, this work sheds light on the genomic basis of cyanide metabolism in the genus *Pseudomonas* and highlights the importance of the in silico comparative genomics for an agile bioprospection of cyanotrophic bacteria and for the identification of novel genes putatively involved in cyanide biodegradation.

## Figures and Tables

**Figure 1 ijms-25-04456-f001:**
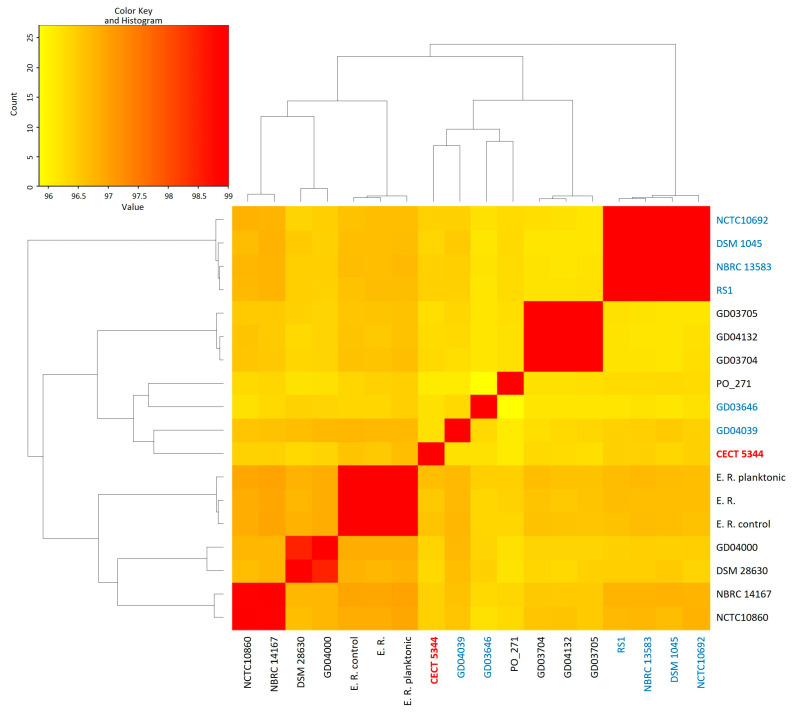
Heatmap and genomic relatedness based on Average Nucleotide Identity (ANI) values obtained for the *P. oleovorans* genomes analyzed in this study. The displayed range of the genomic similarity is given by the color scale. The *nit1C*-containing strains are highlighted.

**Figure 2 ijms-25-04456-f002:**
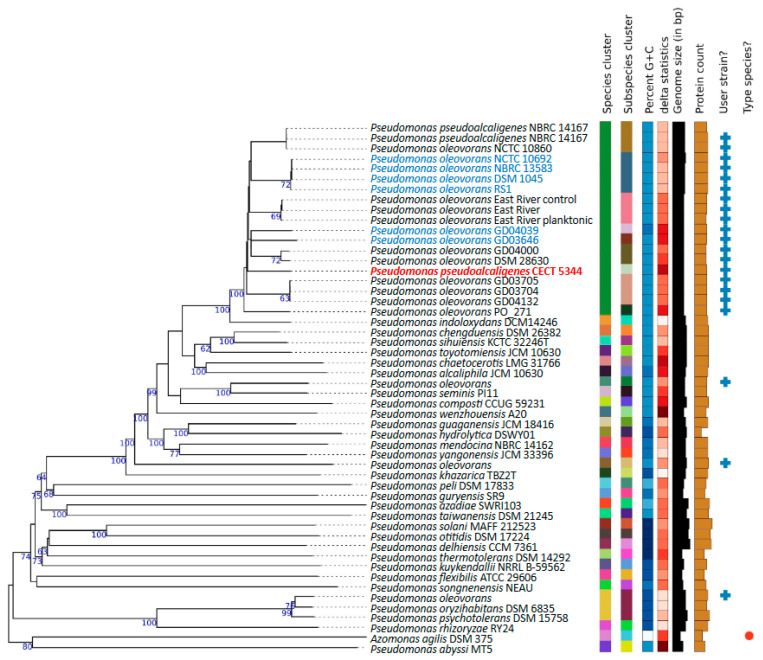
TYGS tree for the *P. oleovorans* species. Tree inferred with FastME 2.1.6.1 from GBDP distances calculated from genome sequences. The branch lengths are scaled in terms of GBDP distance formula d5. The numbers above branches are GBDP pseudo-bootstrap support values > 60% from 100 replications, with an average branch support of 62.8%. The *nit1C*-containing strains are highlighted.

**Figure 3 ijms-25-04456-f003:**
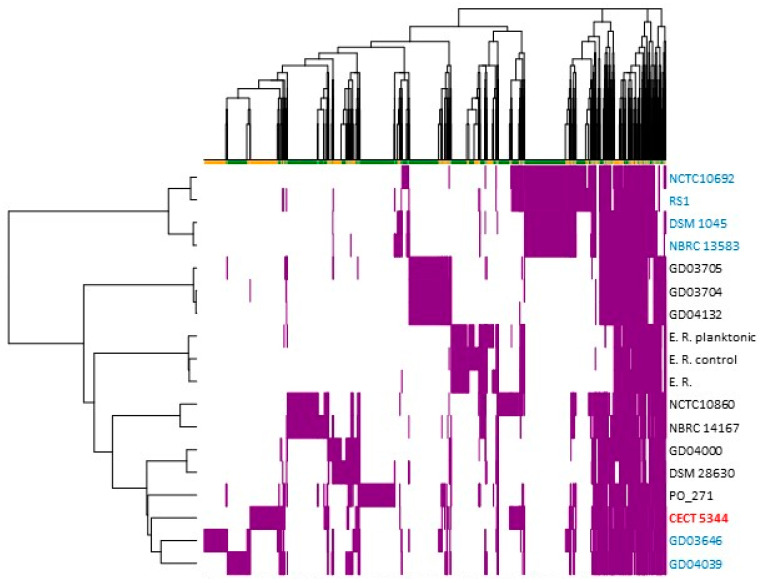
Hierarchical clustering of *P. oleovorans* genomes based on the presence/absence of accessory protein clusters. The *nit1C*-containing strains are highlighted.

**Figure 4 ijms-25-04456-f004:**
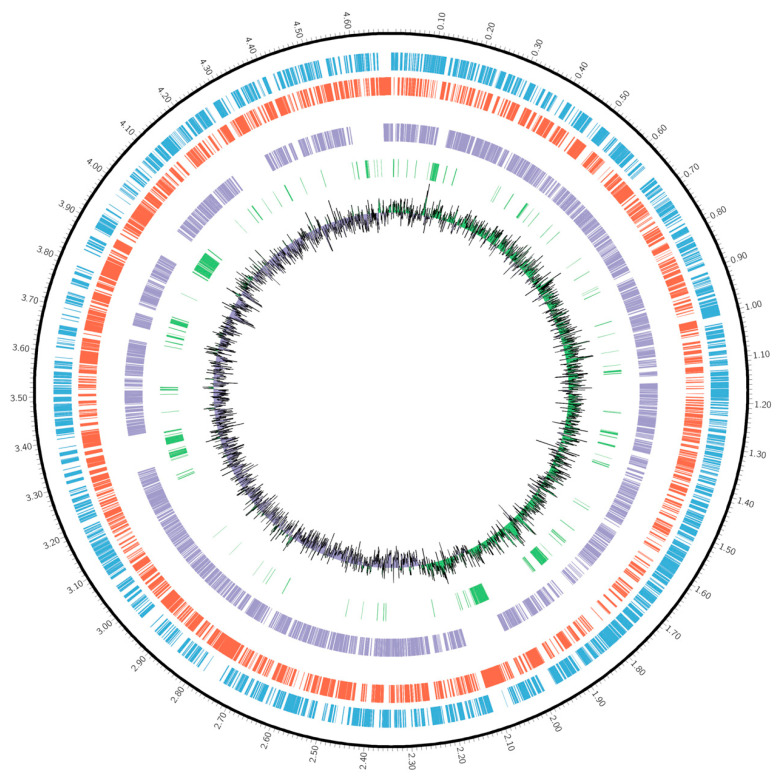
Circos plot of *P. oleovorans* CECT 5344. The different layers represent forward genes (blue), reverse genes (red), core-genes (purple), strain-specific genes (green), and histogram of GC-skew (inner track).

**Figure 5 ijms-25-04456-f005:**
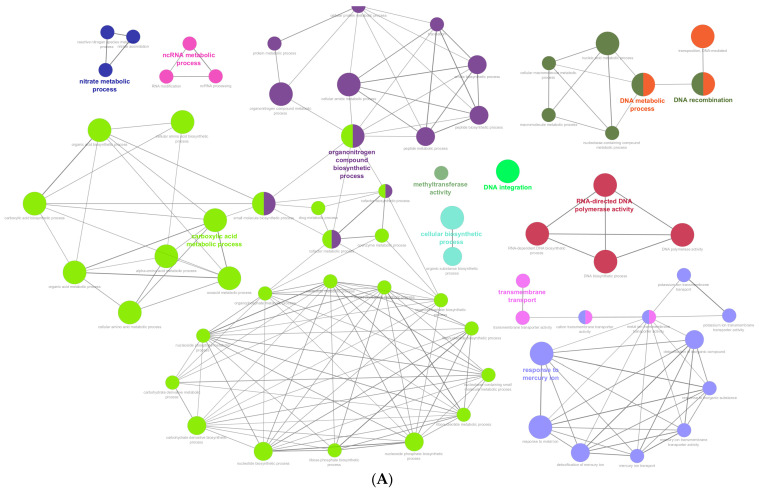
Functional enrichment analysis of the dispensable (**A**) and strain-specific (**B**) genes from *P. oleovorans* CECT 5344. The non-redundant biological function terms were represented as functionally grouped networks. The analysis was performed with the open software platform Cytoscape v3.8 and the tool ClueGO v2.5.10.

**Figure 6 ijms-25-04456-f006:**
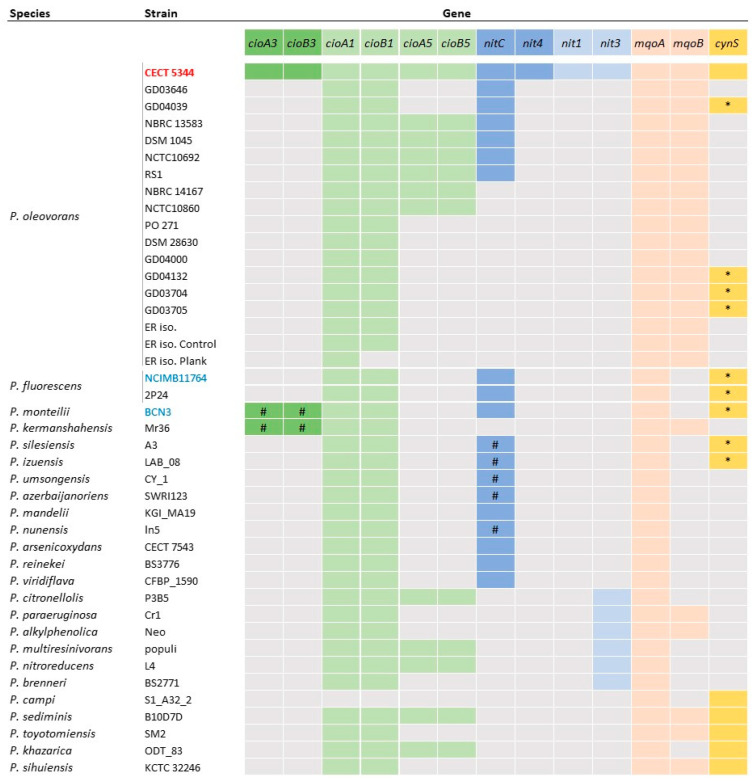
Distribution of genes involved in cyanide metabolism in *Pseudomonas* genomes. Presence or absence of genes is indicated with or without color, respectively. Genes from *P. oleovorans* CECT 5344 whose function have been experimentally demonstrated are indicated with dark colors. Symbols: (#) homologs to the *cioAB* or *nitC* genes located in a different gene cluster when comparing to *P. oleovorans* CECT 5344; (*) cyanases showing low amino acid sequence identity with the CynS from the strain CECT 5344. Cyanotrophic strains are marked in red (CECT 5344) or in blue (NCIMB 11764 and BCN3).

**Table 1 ijms-25-04456-t001:** *Pseudomonas* species and strains containing the *nit1C* gene cluster.

Species	Number of Strains with *nit1C* Cluster	Number of Sequenced Genomes	Relative Abundance (%)
*P. abietaniphila*	1	3	33.3
*P. abyssi*	1	1	100.0
*P. arsenicoxydans*	1	3	33.3
*P. avellanae*	1	18	5.6
*P. bohemica*	1	1	100.0
*P. carbonaria*	1	1	100.0
*P. caspiana*	1	2	50.0
*P. daroniae*	1	4	25.0
*P. fluorescens*	2	299	0.7
*P. folii*	1	1	100.0
*P. gingeri*	1	30	3.3
*P. indoloxydans*	1	2	50.0
*P. kuykendallii*	2	7	28.6
*P. mandelii*	2	13	15.4
*P. migulae*	2	6	33.3
*P. mohnii*	1	3	33.3
*P. monteilii*	1	65	1.5
*P. moorei*	2	5	40.0
*P. oleovorans*	7	36	19.4
*P. quasicaspiana*	1	4	25.0
*P. reinekei*	4	7	57.1
*P. serbica*	1	1	100.0
*P. typographi*	1	3	33.3
*P. viridiflava*	11	1564	0.7
*P. wenzhouensis*	1	2	50.0

## Data Availability

Data is contained within the article and Appendix A.

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
