# Peer review of "Genomic Insights into Cyanide Biodegradation in the Pseudomonas Genus"

_ijms, 2024, doi:10.3390/ijms25084456_

Round 1

Reviewer 1 Report

Comments and Suggestions for Authors

In this article, the authors present an interesting bioinformatic work on phylogenomics of the genus Pseudomonas, as well as a contextualization of different genes involved in cyanide catabolism. It is a shame that the controversy regarding the role of some of the enzymes in this metabolism, as well as the dead-end products generated during it, has not been concluded.

As a comment I suggest to reduce Abstract deleting the sentences "Chemical pollution has become a major environmental concern in recent decades. Bioremediation of contaminants is an eco-friendly technology based on the metabolic versatility of bacteria. This biotechnology has been applied to the decontamination of cyanide-containing industrial wastewaters using cyanotrophic bacteria able to assimilate cyanide". From my point of view, These phrases do not contribute any information to the work shown here.

Line 134. Please give us more information about the number of strains of the genus Pseudomonas and why these were used and not others to obtain the pangenome of the genus. What has been the threshold used to define this pangenome?

Lines 146-147. Please, apport more convincent information about why these 15 Pseudomonas oleovorans genomes were discarded.

Line 165. Please, include some relevant cites, as those from Lalucat's lab.

I also suggest you consider and comment in the Results and Discussion sections on the latest proposal regarding the division of Pseudomonas into several different genera, as well as the phylogenomic studies that support this separation. These studies could better contextualize the analyzes presented here.

Line 195. To obtain the pangenome, only 18 genomes were used. However, for the determination of the ANI, only 21 P. oleovorans were selected. Considering the inclusion of CECT5344, I'm missing some of them. Furthermore, the use of other annotated genomes based on their inclusion in P. pseudoalcaligenes species, as well as other representatives of P. oleovorans, is missed.

Lines 311 and 312. Please a more extent explanation should be included for the well-understanding of the readers.

Lines 325-332. Although the participation of a malate:quinone oxidoreductase as an electron acceptor of the CioAB proteins has been suggested in several articles, the participation of these Mqo proteins should be explained further, highlighting their function as intermediates between the Cio proteins and the transport chain. of electrons in these bacteria. Furthermore, it should be noted that several strains of the genus Pseudomonas (including species other than those explained here) even contain three mqo paralogs, and several of them with a high level of identity/similarity with both MqoA and MqoB of P. pseudoalcaligenes. Please, discuss these points in the manuscript (lines 613-618).

Author Response

Dear Reviewer 1,

First, we would thank all comments and issues raised by the reviewers, which have substantially improved the manuscript after considering in this new version these suggestions and comments. The detailed point-by-point answers to all criticisms, and the changes that we have made in this new version of the manuscript, are described below, but in summary, we have rewritten some parts of the main text to enhance understanding of the manuscript, we have changed Figure 7 by Table 1, which includes additional information, and we have made minor changes to correct some mistakes.

Regards,

Víctor M. Luque-Almagro

Point-by-point answers to the reviewers

Reviewer 1

  1. As a comment I suggest to reduce Abstract deleting the sentences "Chemical pollution has become a major environmental concern in recent decades. Bioremediation of contaminants is an eco-friendly technology based on the metabolic versatility of bacteria. This biotechnology has been applied to the decontamination of cyanide-containing industrial wastewaters using cyanotrophic bacteria able to assimilate cyanide". From my point of view. These phrases do not contribute any information to the work shown here.

The suggested deletion has been made.

  1. Line 134. Please give us more information about the number of strains of the genus Pseudomonas and why these were used and not others to obtain the pangenome of the genus. What has been the threshold used to define this pangenome?

Currently, the genus Pseudomonas comprises 331 validly published species names (LPSN, April 2024). For the selection of the genomes analyzed in this work we used the NCBI Datasets, one of the largest and most comprehensive databases of fully assembled genomes. At the time of carrying out this work, the NCBI Datasets included 31237 genomes belonging to the Pseudomonas genus. Considering that these genomes do not represent uniformly all the species of the genus Pseudomonas (i.e. 30096 genomes belonged to the Pseudomonas aeruginosa species), we decided to use only the high-quality reference genomes corresponding to the genomes representative of the different species from the genus. In addition, these inclusion/exclusion criteria simplified and made faster the analysis. We have included in the new version more information about the selection of genomes used for the determination of the Pseudomonas genus pangenome (lines 731-733).

The threshold used to define the pan-genome of the genus Pseudomonas was the same as used for the pan-genomes of the species P. oleovoras, P. fluorescens or P. monteilii, which consist of a sequence identity level of at least 70% in the BLAST pairwise alignments performed in PanACoTA. Similar thresholds (80%-70%) were applied in previous studies where pangenomes of bacterial genus were determined (Otani et al., 2022, Sci Rep 12, 18909; Caicedo-Montoya et al., 2021, Front. Microbiol. 12:677558). Although more relaxed thresholds have been used in several studies, such as 50% sequence identity and 50% sequence coverage, these relaxed thresholds ended up clustering proteins known to exhibit distinct functions. To avoid that, our stringent threshold at the genus level may fail to cluster some orthologues when only partial conservation is sufficient, a specific BLAST analysis was performed when key genes involved in cyanide metabolism were lost in some genomes. A new sentence clarifying this point has been included in the new version (lines 777-780).

  1. Lines 146-147. Please, apport more convincent information about why these 15 Pseudomonas oleovorans genomes were discarded.

We argue that 15 P. oleovorans genomes were discarded because they were included among the unannotated, redundant, and atypical genomes. Atypical genomes are those for which one or more problems have been identified by NCBI relating to quality, unusual size, or other flaws in the genome assembly. The exclusion of these 15 genomes was also due to the quality control that the PanACoTA software applied, which is focused on basic requirements in terms of quality sequencing of assembly and taxonomy. Specifically, genomes with L90 (the minimum number of contigs necessary to get at least 90% of the whole genome) higher than 100 or genomes with more than 999 contigs, indicative of low quality of sequencing or assembling, resulted in genome exclusion. The second procedure filters redundant and misclassified genomes based on the genetic distance between pairs of genomes, discarding those that showed a Mash distance higher than 0.06. Although these exclusions criteria diminished the number of genomes analyzed, the pan-genomic results obtained were more accurate, reliable, and consistent. This comment is considered in the new version (lines 138-143).

  1. Line 165. Please, include some relevant cites, as those from Lalucat's lab. I also suggest you consider and comment in the Results and Discussion sections on the latest proposal regarding the division of Pseudomonas into several different genera, as well as the phylogenomic studies that support this separation. These studies could better contextualize the analyzes presented here.

We agree with the reviewer about the relevant work performed by the group of Prof. Lalucat on the taxonomy of the genus Pseudomonas, therefore we have included two new references (Lalucat et al., 2022 [33]; Gomila et al., 2022 [34]) that were cited in the Discussion section (lines 458-465).

  1. Line 195. To obtain the pangenome, only 18 genomes were used. However, for the determination of the ANI, only 21 P. oleovorans were selected. Considering the inclusion of CECT5344, I'm missing some of them. Furthermore, the use of other annotated genomes based on their inclusion in P. pseudoalcaligenes species, as well as other representatives of P. oleovorans, is missed.

At the time of carrying out this work, the NCBI Datasets included 36 genomes belonging to the P. oleovorans species. Among these, genomes of the P. pseudoalcaligenes species were also included because according to the new taxonomy molecular tools both species have been considered the same. When the exclusion criteria specified above were applied to these 36 genomes, only 21 genomes met the quality requirements for the determination of the pan-genome of the species P. oleovorans. However, before this, we analyzed if the 21 genomes were correctly assigned to the P. oleovorans species. Taking into account that genomes belonging to a species must have an average nucleotide identity (ANI) higher than 96%, 3 out of 21 genomes showed ANI values <96%, therefore only a total of 18 genomes were used for the construction of the P. oleovorans pan-genome. We have clarified this point in the new version (lines 190-193).

  1. Lines 311 and 312. Please a more extent explanation should be included for the well-understanding of the readers.

With this sentence we want to clarify that homologous to the P. oleovorans CECT 5344 cioA3/cioB3 genes were identified in the strains P. monteilii BCN3 and P. kermanshahensis Mr36, although these genes shown different neighbour genes. The explanation required by the reviewer has been considered in the new version (lines 317-318 and lines 323-324 in the legend of Figure 6).

  1. Lines 325-332. Although the participation of a malate:quinone oxidoreductase as an electron acceptor of the CioAB proteins has been suggested in several articles, the participation of these Mqo proteins should be explained further, highlighting their function as intermediates between the Cio proteins and the transport chain. of electrons in these bacteria. Furthermore, it should be noted that several strains of the genus Pseudomonas (including species other than those explained here) even contain three mqo paralogs, and several of them with a high level of identity/similarity with both MqoA and MqoB of P. pseudoalcaligenes. Please, discuss these points in the manuscript (lines 613-618).

As the reviewer suggested, we have highlighted the function of the Mqo proteins in cyanide resistance and assimilation, and redundant Mqo paralogs in other members of the genus Pseudomonas have been included in the discussion section (lines 628-650).

Reviewer 2 Report

Comments and Suggestions for Authors

A study conducted on “Genomic insights into cyanide biodegradation in the Pseudomonas genus ” is a routine study in environmental microbiology. There are several major points that need restructuring, restating and major language revision.

· In Abstract: the authors stated that:  “Chemical pollution has become a major environmental concern in recent decades. Bioremediation of contaminants is an eco-friendly technology based on the metabolic versatility of bacteria.  The statements should be condensed into one sentence.

· In Abstract: more detailed quantification of key genes in Pseudomonas, to make readers understand and obtain more details.

· Key words should be reduced to five, such as Comparative genomic; cyanide-insensitive oxidase; nitrilase; Pseudomonas; 27 phylogenomics; pan-genome

· In section 1, Line 130-Line 143, the description in this paragraph should be refined, it is too long and vague to understand.

· All the figures are not standardized, and they should be uniformly labeled and expressed in a standardized form, especially in Fig. 3 and Fig. 7.

· In section 2, Line 227, Gene set enrichment analysis (GSA) needs to be improved to quantify which functional genes are upregulated, and which are downregulated.

· Section in Importance of hydrolytic pathways for cyanide biodegradation in bacteria. If there is a hydrolysis pathway diagram, it would be better.

· Conclusions should be refined, it is too long and vague to understand.

Comments on the Quality of English Language

The language of a paper manuscript should be refined during the writing process

Author Response

Dear Reviewer 2,

First, we would thank all comments and issues raised by the reviewers, which have substantially improved the manuscript after considering in this new version these suggestions and comments. The detailed point-by-point answers to all criticisms, and the changes that we have made in this new version of the manuscript, are described below, but in summary, we have rewritten some parts of the main text to enhance understanding of the manuscript, we have changed Figure 7 by Table 1, which includes additional information, and we have made minor changes to correct some mistakes.

Regards,

Víctor M. Luque-Almagro

Point-by-point answers to the reviewers

Reviewer 2

  1. In Abstract: the authors stated that:  “Chemical pollution has become a major environmental concern in recent decades. Bioremediation of contaminants is an eco-friendly technology based on the metabolic versatility of bacteria. ” The statements should be condensed into one sentence.

According to suggestions of reviewer 1, the first sentence of the abstract has been deleted.

  1. In Abstract: more detailed quantification of key genes in Pseudomonas, to make readers understand and obtain more details.

This suggestion has been considered and the second part of the abstract has been rewritten to include the key genes analyzed in this work (lines 16-25).

  1. Key words should be reduced to five, such as Comparative genomic; cyanide-insensitive oxidase; nitrilase; Pseudomonas; 27 phylogenomics; pan-genome

As the reviewer suggested, keywords were reduced to the next five: comparative genomic; cyanide; nitrilase; Pseudomonas; pan-genome.

  1. In section 1, Line 130-Line 143, the description in this paragraph should be refined, it is too long and vague to understand.

This paragraph has been modified according to the comment of the reviewer (lines 128-135).

  1. All the figures are not standardized, and they should be uniformly labeled and expressed in a standardized form, especially in Fig. 3 and Fig. 7.

Figures 1, 2, and 3 were standardized using the same font and highlighting CECT 5344 strain in red and the rest of the strains containing the nit1C gene cluster in blue. Furthermore, thanks to the comment of the reviewer we have considered that Figure 7 can be replaced by Table 1, that provides additional information discussed in the text.

  1. In section 2, Line 227, Gene set enrichment analysis (GSA) needs to be improved to quantify which functional genes are upregulated, and which are downregulated.

In this work, gene set enrichment analysis was performed considering the dispensable and specific genes of the strain CECT 5344. In this sense, genes up- or down-regulated were not presented in this work. By hence, the only quantification we can introduce in this section is the number of dispensable or specific genes that were analyzed by GSA. This quantification was included in line 227.

  1. Section in Importance of hydrolytic pathways for cyanide biodegradation in bacteria. If there is a hydrolysis pathway diagram, it would be better.

We appreciate the suggestion of the reviewer. The hydrolytic reactions catalyzed by NitC and CynD were included in the main text (lines 266-267). In addition, recent references [16,17] in which the hydrolytic pathways for cyanide biodegradation are schematized in a general context have been included.

  1. Conclusions should be refined, it is too long and vague to understand.

The conclusions section has been shortened and modified as suggested by the reviewer.
